# Diet and gut microbiome of skipjack tuna (*Katsuwonus pelamis*) as indicators of environmental changes

Yufei Zhou[1]*, Alejandro Trujillo-González[1], Simon Nicol[1,2], Roger Huerlimann[3], Stephen D. Sarre[1], Dianne Gleeson[1]

1 Centre for Conservation Ecology and Genomics, University of Canberra, Canberra, Australia, 2 Oceanic Fisheries Programme, Pacific Community, Noumea, New Caledonia, 3 Marine Climate Change Unit, Okinawa Institute of Science and Technology Graduate University, Okinawa, Japan

* Yufei.zhou@canberra.edu.au

## Abstract

Understanding the relationship between environmental changes and marine ecosystem dynamics is crucial, especially under the influence of climate events such as the El Niño Southern Oscillation (ENSO). The diet and gut microbiome of marine predators have the potential to efficiently, timely, and reliably indicate impacts of environmental and ecosystem changes, especially with the assistance of high-throughput sequencing (HTS) technology. This study investigated the gut content and microbiome of skipjack tuna (*Katsuwonus pelamis*) collected in the central Pacific Ocean during a transitional period of ENSO phases, shifting from a strong La Niña phase to a weak El Niño phase, aiming to evaluate the impacts of ENSO and other environmental factors on marine food webs and microbiome dynamics of skipjack tuna. While prey diversity was unaffected by ENSO events, skipjack tuna exhibited high diversity and opportunistic foraging patterns, with fish as the primary prey. In contrast, gut microbiome diversity was affected by ENSO events and Southern Oscillation Index (SOI). Five microbiome families (Fusobacteriaceae, Bacillaceae, Propionibacteriaceae, Beijerinckiaceae, and Comamonadaceae), which are associated with immune system functionality and nutritional provisioning of the host, displayed the most significant abundance changes between ENSO phases. A random forest model showed potential for ENSO phase classification based on the abundances of these five families, achieve high accuracy in internal validation, though the performance of external validation was mixed due to storage and sampling period differences. This study highlights the potential of skipjack tuna gut microbiome as indicators of rapid environmental changes, while acknowledging that the short sampling period requires longer-term validation across multiple ENSO cycles.

**Data availability statement:** All sequence data were uploaded to NCBI and accessible with project number PRJNA1242512 (https://www.ncbi.nlm.nih.gov/sra/PRJNA1242512).

**Funding:** This work was supported by a joint scholarship and research fund from the University of Canberra (SCH-u3236264 to YZ) and the European Union Pacific-European-Union-Marine-Partnership Programme granted to the Pacific Community (agreement FED/2018/397-941 to SN), and the New Zealand Government Climate Science for "Ensuring Pacific Tuna Access" project (Ministry of Foreign Affairs and Trade Activity Number: ACT-0103048 to SN). This publication was produced with the financial support of the European Union and Sweden. Its contents are the sole responsibility of the authors and do not necessarily reflect the views of the European Union and Sweden. The funders had no role in study design, data collection and analysis, decision to publish, or preparation of the manuscript.

**Competing interests:** The authors have declared that no competing interests exist.

## Introduction

Marine ecosystems are increasingly threatened by anthropogenic activities, such as fisheries and climate change [1,2]. The complexity of marine ecosystems, both in structure and functionality, makes monitoring and mitigating ecosystem deterioration more challenging. Assessing marine biodiversity is especially difficult because many marine species are inaccessible using traditional methods such as fishing and observation [3,4].

Marine top predators are highly sensitive to ecosystem and environmental changes [5–7]. Unlike species at low trophic levels, top predators are more observable and accessible [8–10]. This makes them excellent indicators of ecosystem dynamics and effects of anthropogenetic disturbances [6,11–13].

Traditional approaches to monitoring marine top predators commonly rely on body condition, morphometrics, or population dynamics to indicate the impacts of environmental or human-induced changes [8,14,15]. However, these methods are less likely to provide insights into ecosystem structure and biodiversity. Additionally, the delayed physiological response of top predators to external pressures [16] complicates their use in real-time ecosystem monitoring.

The gut of marine top predators offers a valuable source of information that can address monitoring challenges. First, gut content or diet analysis provides a great opportunity to assess prey community dynamics [12,17,18] and ecological interactions, such as niche overlap and segregation [19], enabling ecosystem monitoring at the community level. Second, the gut microbiome, which has been associated with the host's nutritional provisioning, metabolism, and immune system functionality [20–25], responds rapidly to environmental and physiological changes. The composition of the gut microbiome and important microbiota can serve as an early indicator of disease, stress, or other health conditions [26–29], making it a reliable tool for assessing top predator fitness in response to external pressures.

Traditional methods of analyzing diet and gut microbiome are limited by low efficiency, the requirement of specialized experts to identify digested prey parts or culture microbiota, and shallow or inaccurate taxonomic identification [30,31]. High-throughput sequencing (HTS) technology significantly improves the efficiency of these analyses by sequence DNA and RNA in high-throughput [31–34]. Meanwhile, HTS enables the analysis of gut contents and microbiome from the same gut lining sample of the host [22,25,35], reducing the need for additional samples and allowing the same gut lining sample to reflect both the diet and the health of the host.

Skipjack tuna (*Katsuwonus pelamis*) (Linnaeus, 1758) is a dominant fishery target in the tropical and subtropical Pacific Ocean. As opportunistic predators, they primarily feed on smaller fish, crustaceans, cephalopods, and mollusks [36,37]. Previous studies suggests that the diet of skipjack tuna is mainly affected by prey availability [38,39] and extreme climate events [25,37]. Therefore, the diet of skipjack tuna has the potential to indicate ecosystem dynamics and external disturbances. However, there is a significant knowledge gap regarding the gut microbiome of skipjack tuna.

Only one study, using historical specimens, showed that El Niño Southern Oscillation (ENSO) events in the Pacific Ocean could affect the diet but not the gut microbiome of skipjack tuna [25].

This study provides a more systematic analysis of the diet and the gut microbiome of freshly caught skipjack tuna during a transitional ENSO period. Gut lining samples were preserved properly and immediately after the catch of the fish to ensure the most accurate representation of the diet and the gut microbiome. A dramatic shift in climate, from a strong La Niña to a weak El Niño phase, occurred within four months between the two sample collection events. These samples allow us to explore whether the diet and the gut microbiome of skipjack tuna can serve as indicators of ecosystem dynamics and environmental changes during a rapid and dramatic climate change event.

## Materials and methods

### Ethics statement

No field access permits were required for this study. All skipjack tuna (*Katsuwonus pelamis*) samples were obtained from commercial purse-seine fishery catches. Gut lining samples were collected during routine fisheries operations coordinated by the Pacific Community. No live animals were handled or subjected to experimental procedures as part of this research. All sampling procedures followed standard commercial fishing practices and regional fisheries management guidelines.

### Sample collection

A total of 15 schools of skipjack tuna located around drifting or anchored fish aggregating devices (FADs) were targeted near Solomon Islands (0539.22S, 15851.447E - 0942.794S, 16118.057E) from December 2022 (school 1–7) to March 2023 (school 8–15) (S1 Fig). Within each school, 10 adult individuals of similar size were captured and processed immediately. Fish sex, body length, and the type of FADs were recorded (S1 File). While this design cannot entirely exclude potential confounding effects, the spatial range is modest relative to skipjack tuna migration patterns and represents regional rather than large scale variation [40,41]. For each fish, the gut lining was extracted using sterilized scissors and forceps, then homogenized with a single-use plastic spoon. Approximately 5 mL of the homogenized gut lining was transferred into 10 mL of RNA*later*® to preserve DNA and RNA. The RNA*later*®-preserved gut lining samples were then frozen at −20°C.

A negative control containing only 10 mL of RNA*later*® was included in each sampling event. The negative control tube remained open from the beginning of the sampling for 30–40 minutes to ensure no cross-contamination during sample processing, after which the tube was capped and stored at −20°C.

The RNA*later*®-preserved samples were transported to the University of Canberra with ice packs within two days in March 2023. Upon arrival, the samples were kept frozen at −20°C before the following molecular analysis.

### Environmental data

Environmental variables (Table 1) used to explore the effects on the gut content and microbiome of skipjack tuna included sea surface temperature (SST), Southern Oscillation Index (SOI), and chlorophyll a concentration (CHLA). These

**Table 1. Environmental variables data sources and details.**

| Variables | Sources | Access date | Unit | Resolution |
|---|---|---|---|---|
| Sea Surface Temperature (SST) | https://neo.gsfc.nasa.gov/view.php?datasetId=MYD28W | July, 2024 | °C | 8-day |
| chlorophyll a concentration (CHLA) | https://neo.gsfc.nasa.gov/view.php?datasetId=MY1DMW_CHLORA | July, 2024 | mg/m$^3$ | 8-day |
| Southern Oscillation Index (SOI) | http://www.bom.gov.au/climate/influences/graphs/#soi-daily | July, 2024 | – | Daily |
| El Niño and the Southern Oscillation (ENSO) | http://www.bom.gov.au/climate/enso/outlook/#tabs=ENSO-Outlook | July, 2024 | – | 14-day |

environmental data were extracted based on the geographic coordinates of each sampling location. Sea surface temperature and CHLA data were collected by National Aeronautics and Space Administration's (NASA) Moderate Resolution Imaging Spectroradiometer (MODIS) sensor on Aqua satellite and obtained via NASA Earth Observations (NEO) platform (https://neo.gsfc.nasa.gov). Data were extracted as 8-day composites corresponding to sampling dates at exact sampling coordinates. El Niño and the Southern Oscillation (ENSO) statuses were determined according to the ENSO outlook from the Bureau of Meteorology, Australian Government. Schools 1–7 of skipjack tuna were collected in December 2022, during the La Niña phase (the peak of the La Niña event). Schools 8–15 were collected in March 2023, during the onset of an El Niño event).

## DNA extraction and amplification

Frozen samples were thawed at room temperature for 20 minutes before DNA extraction. The gut lining samples were diluted by adding an equal volume of ice-cold PBS, following the instructions of the RNA*later*® manufacturer to facilitate effective centrifugation (Thermo Fisher Scientific, Australia). The diluted samples were centrifuged at 7,850 rpm for 10 minutes, after which the supernatant was removed, and the pellet was kept for DNA extraction. DNA extraction was performed using the DNeasy® Blood and Tissue Kit following the protocol provided by the manufacturer (QIAGEN, Germany). The extracted DNA was assessed using a Nanodrop One Spectrophotometer (Thermo Fisher Scientific, Australia) to assess yield and quality, and then stored at −20°C.

## Library preparation and sequencing

For diet analyses, mitochondrial cytochrome c oxidase subunit I region of metazoan species [42,43] was amplified, with a pair of blocking primers to reduce over-amplification of skipjack tuna DNA (Table 2). Each assay contained 2 µL of extracted DNA samples, 1X Gold buffer, 1.5 mM of $MgCl_2$, 0.4 mM of dNTPs, 0.625 U of AmpliTaq Gold Polymerase, 0.3 µM of each primer pair, 0.3 µM of blocking primer, 1/10000 of Sybr Green, and Ultra-purified DNase/RNase-free water to reach a total volume of 25 µL. Each sample was amplified in triplicates using a QuantStudio™ 7 Pro (Thermo Fisher Scientific, Australia) with a touchdown PCR protocol, as recommended by Leray et al. [43], to mitigate non-target amplification. The PCR conditions were: UDG incubation at 50 °C for 2 minutes, initial denaturation at 95 °C for 10 minutes, 16 initial cycles of 95 °C for 10 seconds, annealing at 62 °C for 30 seconds (−1 °C per cycle), and extension at 72 °C for 60 seconds. This was followed by 15 cycles of 95 °C for 10 seconds, annealing at 46 °C for 30 seconds, and extension at 72 °C for 60 seconds. A melt curve was generated from 60 °C to 95 °C with a ramp rate of 0.15 °C per second. Six negative controls were included on each plate. Amplified products were cleaned using AMPure XP beads at a 1:1.2 volume ratio.

Microbiome DNA was amplified using the Pro341F-805R primer set with Illumina overhangs (Table 2) targeting 16S rDNA V3-V4 region of prokaryotes [44]. Assays for microbiome amplification contained 2 µL of extracted DNA samples, 1X Gold buffer, 1.5 mM of $MgCl_2$, 0.4 mM of dNTPs, 0.625 U of AmpliTaq Gold Polymerase, 0.3 µM of each primer pair,

**Table 2. DNA metabarcoding primer sets for metazoan, microbiome, and host-specific blocking.**

| Primer ID | Target taxa | Direction | Sequences (5'-3') | Amplicon size (bp) | Annealing temperature (°C) | Target region | Reference |
|---|---|---|---|---|---|---|---|
| mlCOIintF | Metazoan | Forward | GGWACWGGWTGAACWGTWTAYCCYCC | 313 | 46 | COI | [43] |
| jgHCO2198R | | Reverse | TAIACYTCIGGRTGICCRAARAAYCA | | | | [42] |
| Pro341F | Microbiome | Forward | CCTACGGGNBGCASCAG | 464 | 60 | 16S | [44] |
| Pro805R | | Reverse | GACTACNVGGGTATCTAATCC | | | | |
| SkjBlk1 | Blocking primer | Forward | GGAACAGGTTGAACAGTTTACCCTC CCCTTGCCGG-C3 spacer | – | – | COI | This study |

1/10000 of Sybr Green, and Ultra-purified DNase/RNase-free water to reach a total volume of 25 µL. Each sample was amplified in triplicates using a QuantStudio™ 7 Pro (Thermo Fisher Scientific, Australia) with following PCR conditions: UDG incubation at 50 °C for 2 minutes, initial denaturation at 95 °C for 10 minutes, 33 cycles of 95 °C for 15 seconds, 60 °C for 30 seconds, and 72 °C for 30 seconds. A melt curve was generated from 60 °C to 95 °C with a ramp rate of 0.15 °C per second. Six negative controls and ZymoBIOMICS microbial community standards (in triplicates) as standards were included on each plate. Amplified products were cleaned using AMPure XP beads at a 1:1.2 volume ratio.

Cleaned amplicons were used to construct metabarcoding libraries. First, amplified technical replicates for each sample were pooled and homogenized. Unique indexes were then attached to each sample pool using Illumina DNA/RNA UD indexes (Set A – D) with PCR. Each assay contained 5 µL of cleaned amplicons, 25 µL of Environmental Master Mix 2.0, 10 µL of IDT for Illumina DNA/RNA UD Indexes, and Ultra-purified DNase/RNase-free water to bring the total volume to 50 µL. Indexing PCR was conducted in a QuantStudio™ 7 Pro (Thermo Fisher Scientific, Australia) with the following conditions: initial incubation at 95°C for 10 min, followed by 12 cycles of 95°C for 30 seconds, annealing for 30 seconds, 72°C extension for 30 seconds, and a final extension at 72°C for 5 minutes. The annealing temperature was 46°C for metazoan amplicons and 60°C for microbiome amplicons. Two negative controls were included in each plate. Indexed amplicons were cleaned using AMPure XP beads at a 1:1.2 volume ratio, and the concentration of indexed amplicons was measured with the Qubit HS Assay.

Indexed amplicons were normalized pooled in equal-molar ratios to construct the final libraries. The concentration of final libraries was assessed using the Qubit HS Assay, and the quality and length of the final libraries were assessed on a 1.5% agarose gel. All libraries were sequenced on the Illumina MiSeq platform. For metazoan library, Illumina V3 2x200bp sequencing kit was used. For microbiome library, Illumina V3 2x300bp (600 cycles) sequencing kit was used.

## Bioinformatics analysis

All fastq files generated by Illumina MiSeq were automatically demultiplexed by the Illumina Local Run Manager, with all tags and adapters removed. The quality of the reads was viewed using R (version 4.1.2) [45] and FastQC software [46], followed by primers trimming using cutadapt [47]. The primer-trimmed fastq files were denoised and filtered using *DADA2* (version 1.22.0) [48] under the following conditions: for metazoan reads, forward reads were truncated at 180 bp and reverse read were truncated at 150 bp, and reads with a maximum of two expected errors (MaxEE) were filtered. For microbiome reads, forward reads were truncated at 260, with two MaxEE, while reverse reads were truncated at 220 bp, reads with three MaxEE.

Denoised and filtered forward and reverse reads were merged with 12 bases of overlap, and amplicon sequence variant (ASV) tables without chimeras were constructed using *DADA2* (version 1.22.0) [48].

For metazoan taxonomic assignment, a curated database with COI sequences for metazoan species retrieved from NCBI [49] and BOLD [50] databases was used. For microbiome taxonomic assignment, a curated 16S rDNA V3-V4 region SILVA (138) reference database was used [51]. Taxonomic assignment for both metazoan and microbiome sequences were performed using the naïve Bayesian classifier method [52]. For metazoan assignment, ASVs without phylum-level information and those belonging to the Scombridae family were discarded using the *phyloseq* package (version 1.38.0) [53]. Microbiome ASVs without phylum-level information, or those belonging to chloroplasts or mitochondria, were removed using the *phyloseq* package (version 1.38.0) [53] and the *microbiome* package (version 1.23.1) [54].

## Statistical analysis

All statistical analyses were performed using R (version 4.1.2) [45] within RStudio (version 2022.2.0.443) [55]. Data visualization was conducted using the *ggplot2* package (version 3.4.0) [56].

Prior to alpha diversity analysis, samples were rarefied to equal sequencing depth (5,019 reads per sample for diet data and 4,017 reads per sample for microbiome data) using the *phyloseq* package (version 1.38.0) [51]. Shannon

diversity [57], Chao1 species richness index [58], Simpson's species evenness index [59] were calculated for rarefied diet and gut microbiome data using the *microbiome* package (version 1.23.1) [54]. For the gut microbiome data, the abundance of core microbiome (i.e., ≥ 0.1% relative abundance, > 50% prevalence) was calculated using the *microbiome* package (version 1.23.1) [54]. None of the diversity indices for diet and the gut microbiome followed a normal distribution according to Shapiro-Wilk test [60] (S1 Table). Therefore, the Kruskal-Wallis rank sum test [61] was used to assess the significance of categorical explanatory variables (fish school, fish sex, FADs, catch year) (Table 3) on diversity indices. For continuous explanatory variables (fish length, SST, CHLA, SOI) (**Table 3**), a generalized additive model (GAM) [62] was used to examine the relationship between these variables and diversity indices, using the *mgcv* package (version 1.9.1) [63]. GAMs were fitted using appropriate family distributions based on the data characteristics of each diversity index: gamma family for Chao1 richness and Shannon diversity, and quasi-binomial family for Simpson's evenness index and relative abundance of core microbiome [64]. School was included as a random effect to account for hierarchical structure. Based on diet analysis, fish individuals were grouped into different diet categories (**Table 3**), and gut microbiome diversities were compared among these diet categories using the Kruskal-Wallis rank sum test [61].

Beta diversity was tested using total sum scaling (TSS) normalized data [65]. The gut content and microbiome abundance data were normalized using the *microbiomeMarker* package (version 1.9.0) [66]. Unweighted UniFrac distances were calculated for each sample pair using the *phyloseq* package (version 1.38.0) [53]. Permutational multivariate analysis of variance (PERMANOVA) was then performed with pseudo-F and 999 permutations using the *vegan* package (version 2.6.4) [67] to compare beta diversity differences across explanatory variables. Before interpreting PERMANOVA results, Permutational Multivariate Analysis of Dispersion (PERMDISP) tests were verified using *vegan* package (version 2.6.4) [64] to test for equal multivariate dispersion among groups, treating school as a categorical fixed effect. To verify robustness of results to distance metric choice, we also analyzed beta diversity using centered log-ratio (CLR) transformation followed by Euclidean distance. Beta diversity was visualized using Non-Metric Multidimensional Scaling (NMDS) using the *ape* package (version 5.8) [68].

Diet composition was visualized using the *Metacoder* package (version 0.3.7) [69]. The frequency of occurrence (FOO) and relative abundance (RA) were calculated for each prey family using the *microbiome* package (version 1.23.1) [54] and visualized with a feeding strategy diagram [70]. Relative abundance based on read counts should be interpreted as semi-quantitative due to inherent biases in DNA amplification and sequencing efficiency across taxa. Differential abundance analysis was performed using the *ANCOM-BC2* package (version 2.4.0) [71] to identify significant abundance changes in prey families between ENSO phases.

For microbiome analysis, rare families (prevalence < 1% and detection < 1%) were aggregated into the "other" group, and the log10-transformed abundance of each family in each school was calculated. A heatmap with clustering trees was

Table 3. Category, name, data type and range of explanatory variables.

| Variable category | Variable name | Variable data type | Data range |
|---|---|---|---|
| Fish related variables | School ID | Categorical data | School 1 – School 15 |
| | Sex | Categorical data | Male, Female |
| | Fish length | Continuous data | 38 cm – 56 cm |
| Fishery variables | FADs | Categorical data | Anchored FADs, Drifting FADs |
| Environmental variables | El Niño and the Southern Oscillation (ENSO) | Categorical data | La Niña event, El Niño event |
| | Sea Surface Temperature (SST) | Continuous data | 29.8 °C – 31.4 °C |
| | Southern Oscillation Index (SOI) | Continuous data | −4.2–22.1 |
| | chlorophyll a concentration (CHLA) | Continuous data | 0.07 mg/m$^3$ – 0.67 mg/m$^3$ |
| Diet related variables | Diet type | Categorical data | Fish-diet, Crustacean-diet, Gastropod-diet |
| | Relative abundance of fish in diet | Continuous data | 0.0–1.0 |

plotted to show bacterial family abundance and the similarity among schools using the *aheatmap* function from the *NMF* package (version 0.17.6) [72]. Prior to differential abundance testing, bacterial families were filtered to include only those with ≥ 10% prevalence across samples and ≥ 1% detection threshold. Differential abundance analysis was also performed using the *ANCOM-BC2* package (version 2.4.0) [71] to identify significant changes in gut microbiome families between ENSO phases. Bacterial families showed significant changes between ENSO phases were selected to test the relationship between their log10-transformed abundance and SOI levels using a GAM. A GAM model and a random forest model (RFM) [73] were constructed using the abundance of these bacterial families to predict SOI levels with the *mgcv* package (version 1.9.1) [63] and the *randomForest* package (version 4.7–1.1) [74], respectively. Bacterial family abundances were aggregated at the school level by calculating log10-transformed mean abundances, since SOI values are consistent within each school. The mean squared error (MSE) and R-square ($R^2$) values were compared between the two models. To address potential overfitting of RFM because of the small sample size, we conducted leave-one-school-out internal cross-validation for the RFM. To further test model performance, microbiome data from two previous projects on the gut microbiome of skipjack tuna (GMB [25], TGM [75]) were used as external cross-validation test sets, and the MSE between predicted SOI values by the models and the actual SOI values of each test set were calculated.

## Results

### Diet analysis

After real-time qPCR targeting metazoan species with skipjack-tuna-specific blocking primers, 75 gut samples showed positive amplification and were processed into library preparation stage (5 ± 1 samples per school). The real-time qPCR plates had an average efficiency of 99.6% ± 6% ($R^2$ = 0.984 ± 0.02, Error = 0.232 ± 0.05). All negative controls at each sample processing stage – including sample collection, DNA extraction, and library preparation – showed no or extremely low concentrations of DNA and were excluded from the final Illumina MiSeq sequencing. No additional computational decontamination algorithms were applied beyond negative control screening.

A total of 16,286,678 raw reads were obtained after Illumina MiSeq sequencing (217,155 ± 148,826 reads per sample). After primer trimming, filtering, denoising, and chimera removal, a total of 7,958,010 reads (106,106 ± 61,516 reads per sample) remained. All sequence data were uploaded to NCBI and accessible with project number PRJNA1242512 (https://www.ncbi.nlm.nih.gov/sra/PRJNA1242512).

A total of 180 ASVs were generated and could be unambiguously identified to the phylum level. Only 54 ASVs in 35 genera belonged to non-Scombridae families, of which 44 ASVs belonged to Chordata (all of them were Actinopterygii), 7 ASVs belonged to Arthropoda, 2 ASVs belonged to Mollusca, and 1 ASVs was Platyhelminthes (Fig 1).

A total of 60 gut samples were detected with non-Scombridae metazoan species, and 32 gut samples had only a single prey species detected. None of the explanatory variables significantly affected prey diversity indices (Shannon diversity, Chao1 richness, Simpson's evenness, and beta diversity) (S2 and S3 Table). Most prey detected in the gut of skipjack tuna had low FOO and RA (Fig 2). Skipjack tuna showed a preference for fish prey, as Actinopterygii was detected in almost all gut samples except for four. The most abundant and prevalent fish prey was Myctophidae (FOO = 52%, RA = 30%) and Acanthuridae (FOO = 24%, RA = 14%). Gastropoda, represented by the Cavoliniidae family, also showed importance in the gut of skipjack tuna (FOO = 26%, RA = 12%). Malacostraca was detected in eight gut samples, with the Trapeziidae family being the most important crustacean family (FOO = 9%, RA = 8%). Differential abundance analysis showed no significant abundance changes in any prey families of skipjack tuna between ENSO phases.

### Gut microbiome analysis

Based on data quality, a total of 142 samples were selected for sequencing (9 ± 1 samples per school). Real-time qPCR had an average efficiency of 92.4% ± 5% ($R^2$ = 0.954 ± 0.03, Error = 0.289 ± 0.11). All negative controls during sample

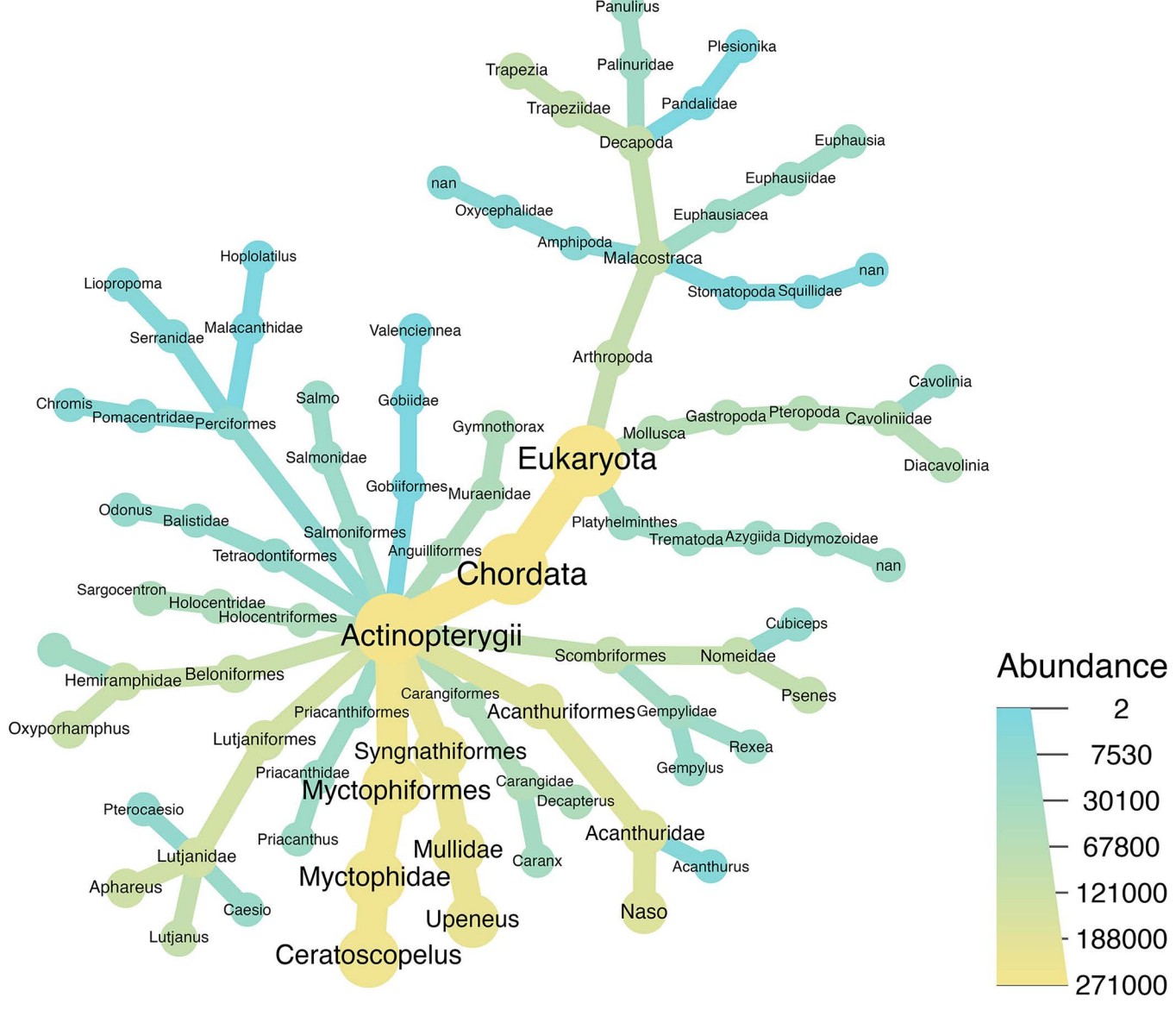

**Fig 1. Prey composition and number of ASVs detected in the gut of skipjack tuna.** Colored by the read abundances.

processing – including sample collection, DNA extraction, and library preparation – showed no or extremely low concentration of DNA and were excluded from the final Illumina MiSeq sequencing.

A total of 5,461,861 raw reads were obtained after Illumina MiSeq sequencing (38,737 ± 18,059 reads per sample). After primer trimming, filtering, denoising, and chimera removal, 2,914,195 reads (20,522 ± 4,730 reads per sample) remained. A total of 1,784 ASVs were generated, all of which could be unambiguously identified to the phylum level and were not assigned to chloroplast and mitochondria. All sequence data were uploaded to NCBI and accessible with project number PRJNA1242512 (https://www.ncbi.nlm.nih.gov/sra/PRJNA1242512).

No explanatory variable significantly affected Shannon diversity (S4 and S5 Tables). The abundance of the core microbiome, Chao1 richness, and Simpson's evenness were significantly different among schools (Kruskal-Wallis rank sum

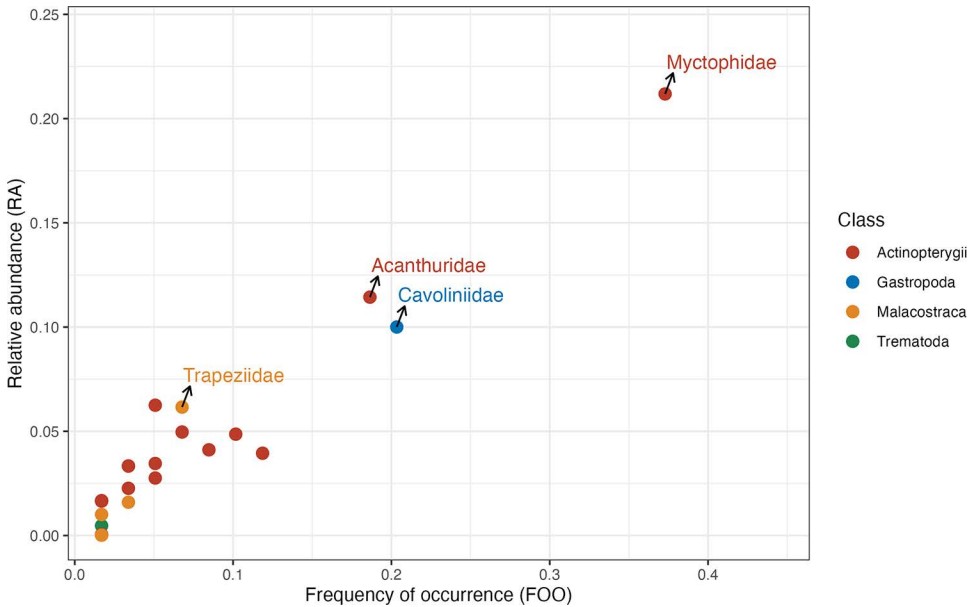

**Fig 2. Foraging pattern of skipjack tuna.** Each point indicates a prey family. The x-axis indicates the frequency of occurrence (FOO) of the prey family across all gut samples, the y-axis indicates the relative abundance (RA) of the prey family across all gut samples. Color of the points indicates the class of the prey. The most important families in Actinopterygii, Gastropoda, and Malacostraca are labelled.

test: core microbiome abundance: $\chi^2$ (14) = 32.5, $p < 0.01$; Chao1: $\chi^2$ (14) = 33.7, $p < 0.01$; Simpson's evenness: $\chi^2$ (14) = 26.1, $p = 0.03$). Only Simpson's species evenness showed significant difference between ENSO phases (Kruskal-Wallis rank sum test: $\chi^2$ (1) = 4.86, $p = 0.03$). Chao1 richness and Simpson's evenness showed significant associations with SOI when analyzed using GAMs with appropriate family distributions (Chao1: Gamma family: $R^2 = 0.004$, $t = -1.93$, $p = 0.05$; Simpson's evenness: quasi-binomial family: $R^2 = 0.005$, $t = 2.33$, $p = 0.02$) (Fig 3A). School effects were insignificant as random effects in both models. No other significant effects were observed between diversity indices and explanatory variables (S4 and S5 Table). Beta diversity of the gut microbiome was significantly affected by school (PERMANOVA test: $F_{(14, 127)} = 1.67$, $R^2 = 0.16$, $Pr < 0.01$,), ENSO (PERMANOVA test: $F_{(1, 140)} = 3.03$, $R^2 = 0.02$, $Pr < 0.01$), SOI (PERMANOVA test: $F_{(1, 140)} = 2.93$, $R^2 = 0.02$, $Pr < 0.001$), FADs (PERMANOVA test: $F_{(1, 140)} = 2.77$, $R^2 = 0.02$, $Pr < 0.01$), and CHLA (PERMANOVA test: $F_{(1, 140)} = 2.81$, $R^2 = 0.02$, $Pr < 0.001$) (Fig 3B). PERMDISP tests confirmed equal multivariate dispersion for the primary variables of ENSO, SOI and school, but not for FADs and CHLA (S6 Table). To ensure robustness across analytical approaches, we also complemented unweighted UniFrac analysis with CLR transformation and Euclidean distance, which yielded consistent results (S6 Table).

Vibrionaceae was the most abundant microbiome family across all schools, indicating a nearly fresh status of the gut microbiome samples from skipjack tuna [75] (Fig 4). Based on the abundances of bacterial families, schools collected during the same ENSO phase clustered together, with the exception of school 1 and school 9 (Fig 4).

Five families showed significant abundance changes between ENSO events according to the ANCOM-BC2 differential abundance analysis (all $p < 0.001$ and $q < 0.001$). Fusobacteriaceae abundance significantly increased during the transition from La Niña to El Niño phase (LFC = 1.10, $q < 0.001$), while the abundances of Bacillaceae (LFC = 0.78, $q < 0.001$), Propionibacteriaceae (LFC = 0.76, $q < 0.001$), Beijerinckiaceae (LFC = 1.29, 95% CI: 0.51 to 2,06, $q < 0.001$), and Comamonadaceae (LFC = 1.52, 95% CI: 0.69 to 2.35, $q < 0.001$) significantly decreased during the transition phase (Fig 5A). Three families (Bacillaceae, Propionibacteriaceae, and Fusobacteriaceae) showed complete separation between ENSO phases (SE = 0).

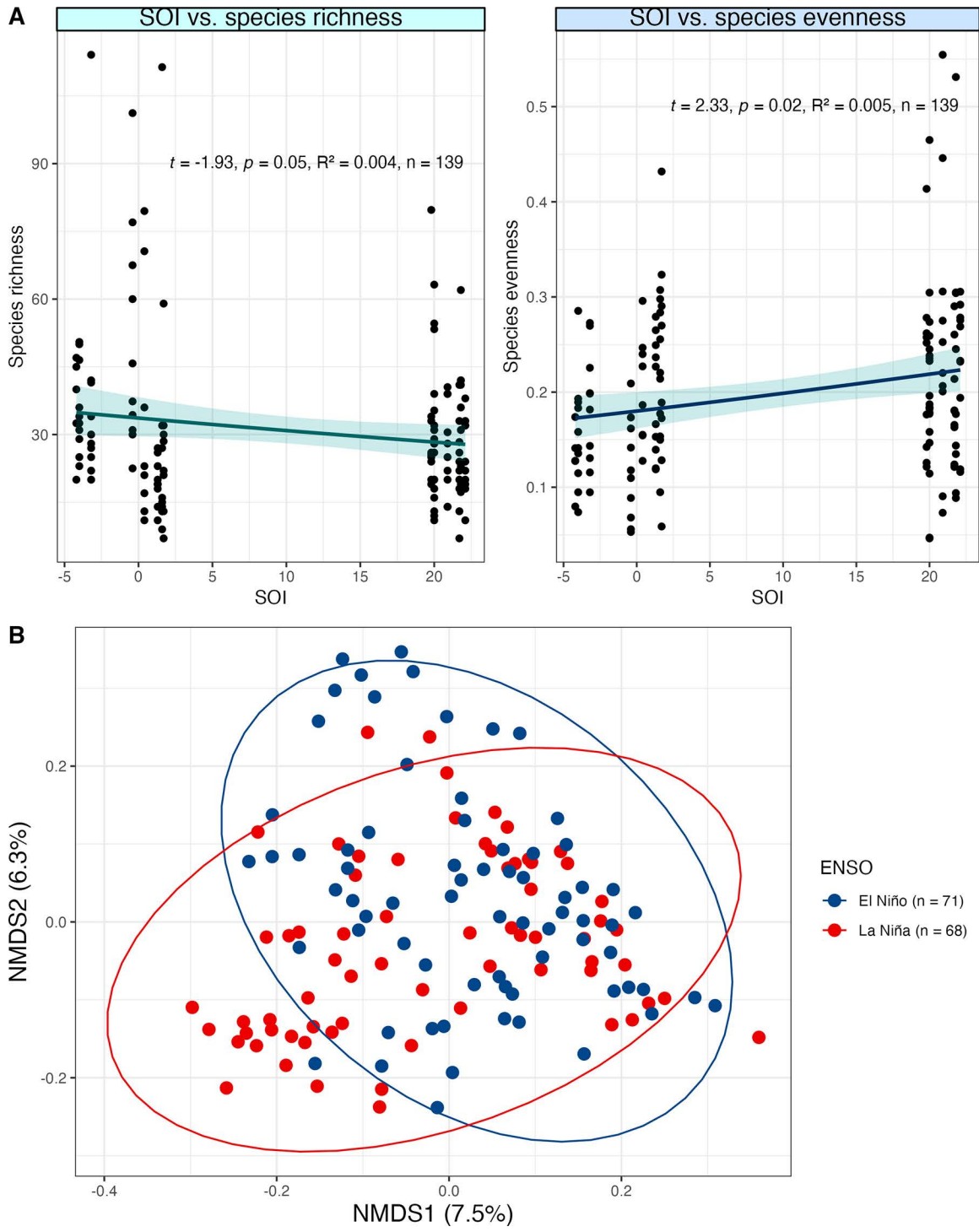

**Fig 3. Alpha and beta diversities of the gut microbiome of skipjack tuna. (A)** GAM relationships between diversity indices and SOI using appropriate family distributions (Chao1: Gamma family; Simpson's evenness: quasi-binomial family). Shaded areas represent 95% confidence intervals, with statistical results annotated. **(B)** Beta diversity NMDS plot, colored by ENSO events (red: La Niña event; blue: El Niño event) with 95% confidence ellipses.

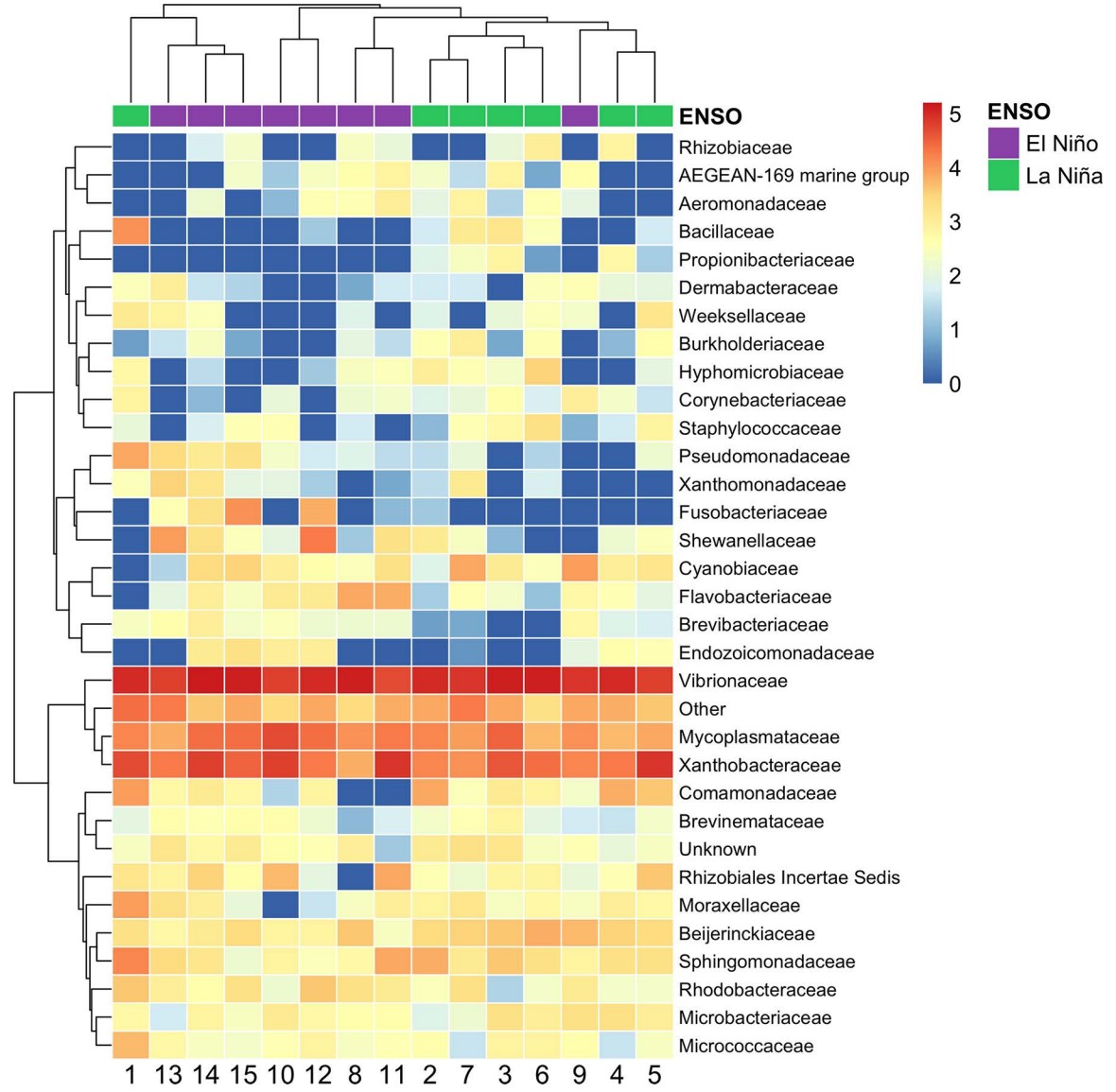

**Fig 4. The gut microbiome family-level abundance in skipjack tuna schools.** The abundances of each family were log10-transformed. Schools from different ENSO events were annotated above the heatmap. Rare families (prevalence < 1% and detection < 1%) were grouped into "Other" group.

The generalized additive models showed non-linear relationships between the abundances of the significantly changed families and SOI, especially in Comamonadaceae, Beijerinckiaceae, and Propionibacteriaceae (Fig 5B). A GAM and a RFM were constructed and compared, using the school-level log10-transformed abundance of the five families as predictor variables and SOI as the predicted variable. Overall, the RFM outperformed the GAM, showing lower mean squared error (MSE) and higher explained variances ($R^2$) (Table 4). While the internal cross-validation for RFM showed moderate precision ($R^2 = 0.90$, RMSE = 5.2), it achieved perfect ENSO phase classification in this proof-of-concept analysis (100% accuracy, n = 15 schools) using the SOI threshold of 7 (S7 Table). External cross-validation with test sets indicated that the RFM provided a more accurate SOI prediction for TGM data but not for GMB data (Table 4). The GAM also showed a prediction with high mean squared error for GMB data and its performance on TGM data was less accurate than RFM.

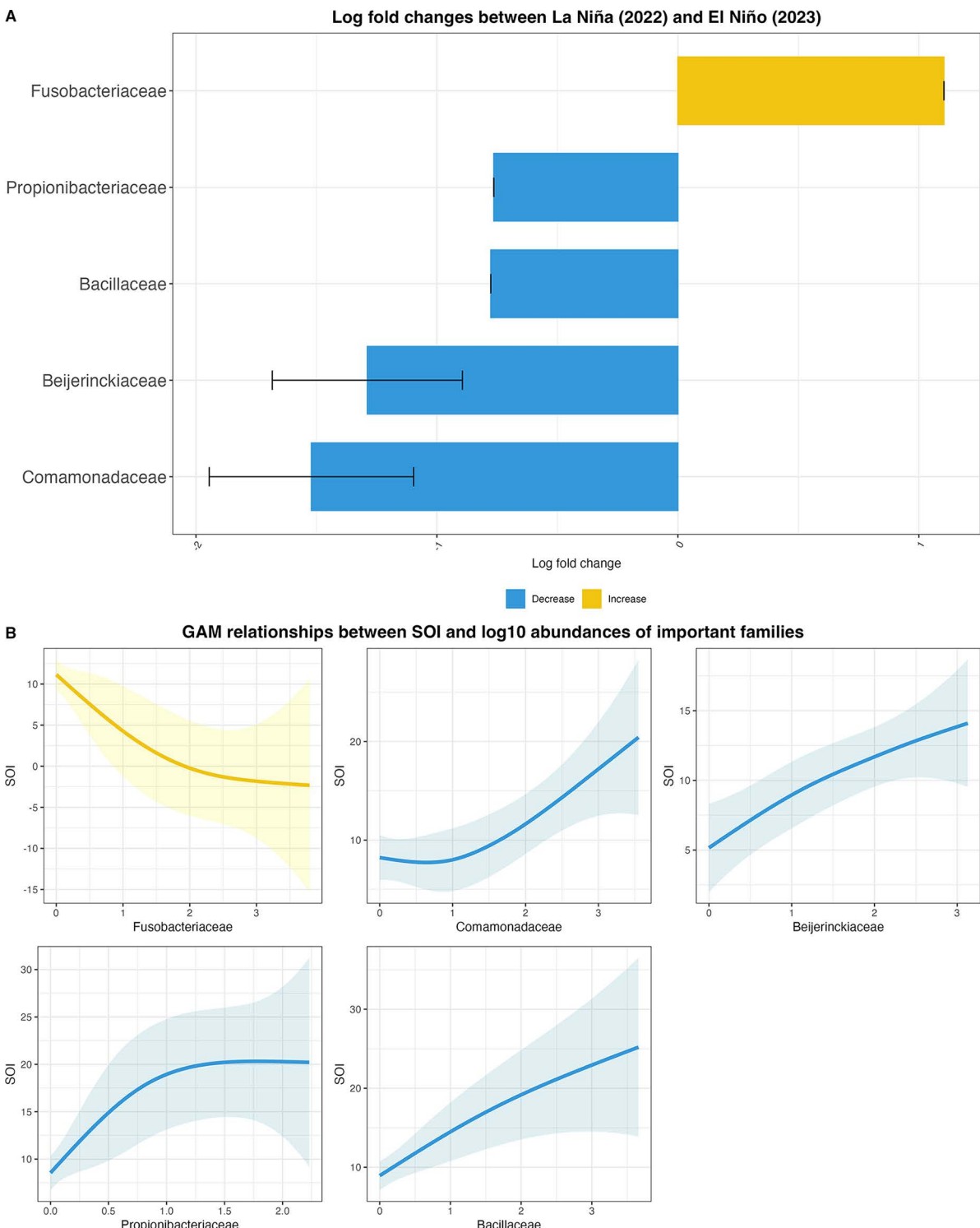

**Fig 5. Five significantly changed gut microbiome families between ENSO events and with SOI change. (A)** ANCOM-BC differential abundance analysis result indicates the significant abundance changes in five families between El Niño and La Niña event. Analysis included bacterial families with ≥10% prevalence and ≥1% detection threshold; (**B**) GAM relationships between log10 abundances of the five families and SOI. Colored by the change direction (yellow: the abundance increased from La Niña to El Niño, blue: the abundance decreased from La Niña to El Niño).

**Table 4. Generalized additive model (GAM) and random forest model (RFM) using school-level log10-transformed abundances of Fusobacteriaceae, Bacillaceae, Propionibacteriaceae, Beijerinckiaceae, Comamonadaceae to predict Southern Oscillation Index (SOI). Here shows the mean squared error (MSE) and R-square (R²). External validations were conducted on independent datasets (GMB and TGM). MSE between predicted and actual SOI for two test datasets were shown.**

| Model name | Predictor variables | Predicted variable | MSE | R² | MSE between predicted and actual SOI for GMB data [25] | MSE between predicted and actual SOI for TGM data [75] |
|---|---|---|---|---|---|---|
| GAM | Log10 abundances of Fusobacteriaceae, Bacillaceae, Propionibacteriaceae, Beijerinckiaceae, and Comamonadaceae | SOI | 12.8 | 0.895 | 267.7 | 33.2 |
| RFM | | | 4.9 | 0.991 | 253.9 | 0.1 |

Permutation importance analysis identified Propionibacteriaceae and Bacillaceae as the most important predictors of SOI in the random forest model (S2 Fig). Partial dependence plots revealed that Fusobacteriaceae showed a negative relationship with predicted SOI, while the other four families (Comamonadaceae, Beijerinckiaceae, Propionibacteriaceae, and Bacillaceae) exhibited positive relationships with SOI predictions (S3 Fig).

## Discussion

This study investigated the potential of skipjack tuna diet and gut microbiome as indicators of environmental and ecosystem change during the rapid and dramatic transitional period from a La Niña event (2022) to an El Niño phases (2023). While the diet showed no significant variation, the gut microbiome exhibited strong correlation with ENSO events and SOI. Notably, five bacterial families were identified as key predictors of SOI levels using a random forest model, highlighting the gut microbiome's potential as a sensitive and accurate indicator of climate changes.

### Diet analysis

This study demonstrated the feasibility of using high-throughput sequencing (HTS) technology to assess the diet of skipjack tuna. However, non-Scombridae species were successfully amplified and sequenced in only 60 gut samples (40%), with over half of those detecting just one species. Given the high frequency of empty stomachs in skipjack tuna [38,76], the low concentration and highly digested prey DNA, and the over-represented host DNA are likely to affect the performance of DNA metabarcoding. Although we employed host-blocking primers to reduce skipjack tuna DNA amplification, future studies could further improve detection rates through alternative genetic markers targeting different genes with varying degradation rates, increased sequencing depth to enhance detection of rare prey DNA, or multi-marker approaches to improve taxonomic coverage [77,78]. Nevertheless, 34 genera were detected in this study, which still surpasses previous diet analyses using isotopic or morphological identification methods in terms of diversity and taxonomic resolution [38,79].

Our results support that skipjack tuna are opportunistic predators, feeding on a wide range of prey with low abundance and prevalence [70]. Actinopterygii was the most abundant and diverse taxon, with common prey families such as Myctophidae (lantern fish), Acanthuridae (unicorn fishes, surgeonfishes, and tangs), Gempylidae (snake mackerels), Lutjanidae (fusiliers), Serranidae (groupers), consistent with previous morphological and isotopic studies [37,38,80]. Crustaceans also played an important role, as reported in prior studies [76,79]. Interestingly, Cavoliniidae, a gastropod family previously undetected in skipjack tuna diets, showed importance in this study. As pelagic molluscs in tropical waters, Cavoliniidae larvae are an important prey of higher predators [81], suggesting the utility of HTS to detect less-observable prey.

No significant differences in diet were observed across fish-related, fishery, or environmental variables, potentially due to the limited samples size and the short study timeframe of four months, which may be insufficient to capture ecosystem responses, as time lags between environmental and ecosystem changes can span months to years [16].

## Gut microbiome analysis

**Changes in microbiome diversity.** In contrast to diet, the gut microbiome of skipjack tuna showed detectable response to environmental changes during the study period. Significant differences in gut microbiome richness, evenness, and beta diversity were observed between ENSO phases and SOI levels. During La Niña phase, skipjack tuna had a less diverse and more even gut microbiome, highlighting the potential of skipjack tuna gut microbiome as indicators of ENSO effects. Several studies have proposed using fish gut microbiome to indicate environmental pressures [82–84]. This study may be the first to document detectable ENSO-associated changes in a marine fish gut microbiome over a four-month transitional period. Other factors, including sea surface temperature, chlorophyll-a concentration, and diet, did not significantly influence the gut microbiome diversity of skipjack tuna, further emphasizing the role of ENSO and SOI in shaping the gut microbiome communities. While ENSO and SOI explained a relatively small proportion of total variance in microbiome composition, such modest effect sizes are typical in ecological microbiome studies and can still reflect ecologically important responses to environmental change.

The gut microbiome composition of skipjack tuna in this study aligns with a previous study [75]. As reported, high abundances of Vibrionaceae and Mycoplasmataceae indicate reliable gut sample preservation, confirming that the gut samples in this study can reliably represent the "true" gut microbiome of living fish.

Spatial confounding might be less significant because the 4–5 degree latitudinal sampling range represents a modest variation relative to skipjack tuna migration patterns and regional stock distribution [40,41,85]. However, we acknowledge that our two-timepoint design cannot definitively separate ENSO effects from seasonal variations. Systematic sampling across multiple ENSO cycles at fixed locations would be required to conclusively separate ENSO effects from temporal and spatial variation.

**Changes in important bacterial families.** Five bacterial families exhibited significant abundance changes between La Niña and El Niño phases. Fusobacteriaceae is the only family showed a low abundance during the La Niña phase (i.e., abundance increased during the El Niño phase). Fusobacteriaceae commonly shows significance when the living-condition change for the host fish, such as contaminations [86], change of culture mode [87], or change of feeding strategy [88]. This family is also closely related to immune system functionality [89]. This suggests that the rapid transition from La Niña to El Niño conditions may have triggered immune responses in skipjack tuna.

The families Comamonadaceae, Bacillaceae, Propionibacteriaceae, and Beijerinckiaceae were more abundant during La Niña. These families have been associated with probiotic functions in metabolism, digestion, and nutritional provisioning in previous studies [90–95]. For example, species within Bacillaceae (e.g., *Bacillus subtilis*) are common probiotic additives in aquaculture feeds to improve fish digestion and immunity [90,93]. Genera within Beijerinckiaceae (e.g., *Bosea*) and Propionibacteriaceae (e.g., *Propionibacterium* and *Propioniciclava*) are closely related to high lipid and high protein feeding strategy in fish [91,92,94,96]. Comamonadaceae, in particular, is known to decline under stress [97,98]. The decrease in these probiotic families during the transition to El Niño suggests that rapid environmental changes placed potential physiological stress on skipjack tuna, affecting their digestion and immunity.

However, taxonomic identification at the family level has limitations for inferring function. Bacterial families are ecologically diverse, and functional capabilities can vary substantially among genera and species within families [99–101]. Therefore, the proposed associations between specific families and probiotic or immune-related functions should be considered hypotheses requiring validation rather than established mechanisms. Future studies employing metagenomics and metatranscriptomics would enable direct assessment of functional gene content and expression, providing insights into how these bacterial communities contribute to host metabolism, immunity, and stress responses during ENSO events.

**Bacterial families as ENSO phase indicators.** The abundances of the five bacterial families demonstrated potential for ENSO monitoring using a random forest model (RFM), but this must be interpreted cautiously given the small sample size. In this proof-of-concept study, internal cross-validation revealed that while SOI prediction showed moderate precision, the model achieved perfect accuracy in classifying ENSO phases. Similar classification performance was

observed in external validation using TGM dataset, suggesting that this exploratory approach has the potential for further investigation with larger sample size across complete ENSO cycles.

However, both RFM and GAM showed low accuracy for the GMB test data, highlighting limitations in broader application of the model. The poor external validation performance could be related to two factors. First, SOI range mismatch: a large proportion of skipjack tuna samples in the GMB project were collected during peak El Niño phases (SOI: −20.2 to 9.7), whereas this study's SOI range was −4.2 to 22.1. The lack of peak El Niño data in this study likely limited the model's broader applicability, particularly its ability to predict SOI in El Niño phases. Second, preservation and storage effects: GMB gut samples had been stored for a long period with different preservation protocols compared to our freshly collected, immediately preserved samples. Microbiome composition can shift over time [75], potentially altering the bacterial family abundance patterns and affecting predictive accuracy.

This work represents proof-of-concept that gut microbiome composition and five bacterial families have the potential to distinguish ENSO phases in skipjack tuna. Future improvements to predictive models would benefit from systematic sampling across complete ENSO cycles to capture the full range of environmental conditions, particularly peak El Niño phases, larger sample sizes spanning multiple years, and standardized preservation protocols to ensure microbiome stability. Such comprehensive approaches would strengthen further operational deployment of microbiome-based environmental monitoring tools.

## Conclusions

This study examined the gut content and microbiome of skipjack tuna to assess their potential as indicators of ecosystem dynamics and environmental changes.

High-throughput sequencing proved to be a valuable tool for assessing ecosystem structure through the diets of opportunistic predators. While skipjack tuna exhibited a diverse and opportunistic diet, diet results should be considered preliminary given the moderate prey detection rate. No significant relationship was found between diet composition and ENSO events potentially due to the high empty stomach rate, limited sample size, and the short sampling timeframe.

The gut microbiome showed detectable changes associated with ENSO events and SOI level, with significant changes in species richness and evenness. The abundances of five key bacterial families (Fusobacteriaceae, Bacillaceae, Propionibacteriaceae, Beijerinckiaceae, and Comamonadaceae) were significantly related to ENSO fluctuations, with predictive potential demonstrated using a random forest model. These findings suggest that extreme climate events can rapidly influence skipjack tuna, with changes in gut microbiome composition and changes in bacterial families associated with immune and nutritional functions.

This proof-of-concept study demonstrates that skipjack tuna gut microbiome composition responds rapidly to ENSO transitions and can distinguish between climate phases. However, the findings are constrained by the short sampling period and small sample size, which cannot definitively separate ENSO effects from seasonal or spatial variation. To expand on this foundational work, a systematic, long-term study is needed to better understand the links between environmental changes, diet, and gut microbiome dynamics in skipjack tuna. Such research will be crucial for integrating microbiome-based insights into marine ecosystem monitoring and management.

## Supporting information

**S1 Fig. Map of sampling locations.** Red points indicate samples collected in 2022 (school 1–7) and blue points indicate samples collected in 2023 (school 8–15).
(DOCX)

**S1 File. Metadata sheet of skipjack tuna and environmental variables.**
(CSV)

 

**S1 Table. Results for Shapiro-Wilk normality test for alpha diversity indices.**
(DOCX)

**S2 Table. Kruskal-Wallis rank sum test for the association between diet diversity of skipjack tuna and categorical explanatory variables.**
(DOCX)

**S3 Table. GAM test for the association between diet diversity of skipjack tuna and continuous explanatory variables.**
(DOCX)

**S4 Table. Kruskal-Wallis rank sum test for the association between gut microbiome diversity of skipjack tuna and categorical explanatory variables.**
(DOCX)

**S5 Table. GAM test for the association between gut microbiome diversity of skipjack tuna and continuous explanatory variables.**
(DOCX)

**S6 Table. PERMANOVA result for beta diversity of gut microbiome of skipjack tuna in association with explanatory variables.**
(DOCX)

**S2 Fig. Variable importance in random forest model (RFM) for SOI prediction.** Permutation importance scores indicate the relative contribution of each bacterial family to model performance.
(DOCX)

**S3 Fig. Partial dependence plots showing relationships between bacterial family abundances and predicted SOI values.** Each panel shows the marginal effect of one bacterial family's log10-transformed abundance (x-axis) on predicted SOI (y-axis) while averaging over the effects of other families in the random forest model.
(DOCX)

**S7 Table. Leave one school out internal test result for random forest model (RFM).** Abundances of five important families (Fusobacteriaceae, Bacillaceae, Propionibacteriaceae, Beijerinckiaceae, and Comamonadaceae) are explanatory variables, and ENSO event is the prediction variable.
(DOCX)

## Acknowledgments

We thank Aurelie Guillou, Bruno Leroy, and Joe Scutt Phillips and other technical specialists from the Pacific Community for their assistance in sample collection and transportation. We thank Jenn Soroka, Kymberly Crockett, and Henriette Theron from the EcoDNA group for their assistance in lab works.

## Author contributions

**Conceptualization:** Alejandro Trujillo-González, Simon Nicol, Dianne Gleeson.

**Data curation:** Yufei Zhou, Alejandro Trujillo-González, Roger Huerlimann.

**Formal analysis:** Yufei Zhou.

**Funding acquisition:** Alejandro Trujillo-González, Simon Nicol, Dianne Gleeson.

**Investigation:** Yufei Zhou.

**Methodology:** Yufei Zhou, Alejandro Trujillo-González, Simon Nicol, Roger Huerlimann, Stephen D. Sarre, Dianne Gleeson.

**Project administration:** Alejandro Trujillo-González, Simon Nicol, Dianne Gleeson.

**Resources:** Alejandro Trujillo-González, Simon Nicol, Dianne Gleeson.

**Software:** Roger Huerlimann.

**Supervision:** Alejandro Trujillo-González, Simon Nicol, Roger Huerlimann, Stephen D. Sarre, Dianne Gleeson.

**Validation:** Alejandro Trujillo-González, Simon Nicol, Roger Huerlimann, Dianne Gleeson.

**Visualization:** Yufei Zhou.

**Writing – original draft:** Yufei Zhou.

**Writing – review & editing:** Alejandro Trujillo-González, Simon Nicol, Roger Huerlimann, Stephen D. Sarre, Dianne Gleeson.

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
