## [Decision Letter · Decision Letter 0]

20 Sep 2025

Dear Dr. Zhou,

Thank you for submitting your manuscript to PLOS ONE. After careful consideration, we feel that it has merit but does not fully meet PLOS ONE’s publication criteria as it currently stands. Therefore, we invite you to submit a revised version of the manuscript that addresses the points raised during the review process.

We look forward to receiving your revised manuscript.

Kind regards,

Sandipan Mondal, Ph.D.

Academic Editor

PLOS ONE

Journal Requirements:

This work was supported by the University of Canberra and by the European Union “Pacific-European-Union-Marine-Partnership” Programme (agreement FED/2018/397-941) grant to the Pacific Community. This publication was produced with the financial support of the European Union and Sweden. Its contents are the sole responsibility of the authors and do not necessarily reflect the views of the European Union and Sweden.

5. Please remove all personal information, ensure that the data shared are in accordance with participant consent, and re-upload a fully anonymized data set.

Additional guidance on preparing raw data for publication can be found in our Data Policy (https://journals.plos.org/plosone/s/data-availability#loc-human-research-participant-data-and-other-sensitive-data) and in the following article: http://www.bmj.com/content/340/bmj.c181.long....

Reviewers' comments:

Reviewer's Responses to Questions

**Comments to the Author**

1. Is the manuscript technically sound, and do the data support the conclusions?

Reviewer #1: Partly

Reviewer #2: Yes

2. Has the statistical analysis been performed appropriately and rigorously?

Reviewer #1: Yes

Reviewer #2: Yes

3. Have the authors made all data underlying the findings in their manuscript fully available?

Reviewer #1: No

Reviewer #2: Yes

4. Is the manuscript presented in an intelligible fashion and written in standard English?

Reviewer #1: Yes

Reviewer #2: Yes

Reviewer #1: Diet and gut microbiome of skipjack tuna (Katsuwonus pelamis) as indicators of environmental changes

Manuscript Number: PONE-D-25-46392

Summary & Overall Assessment

This manuscript examines whether the diet and gut microbiome of skipjack tuna can act as indicators of rapid environmental change across a four-month transition from La Niña (Dec 2022) to the onset of El Niño (Mar 2023) near the Solomon Islands. The study sampled 15 tuna schools (∼10 fish per school), preserved gut lining for both diet metabarcoding (COI) and 16S rRNA microbiome profiling, and related community patterns to ENSO phase, SOI, SST, and chlorophyll-a. Diet composition showed no significant ENSO signal, whereas the microbiome displayed significant shifts in alpha and beta diversity, with five bacterial families differing between phases and supporting SOI prediction via a random-forest model.

The work is timely and potentially useful to ecosystem monitoring. Methodologically, it is generally careful (negative controls, standards, dual assays, transparent pipelines). However, several analytical and reporting issues need attention—especially study design/replication, statistical handling (PERMANOVA assumptions, alpha-diversity modeling), model validation for SOI prediction with very small training sets, and fuller ethics/data/code reporting. I recommend major revision.

Title and Abstract

Strengths: Clear, accurate, and aligned with results; avoids overstating novelty. The abstract correctly contrasts weak diet signals with stronger microbiome responses, and it names the five key families and the modeling approach.

Suggestions:

1. Tone down “successfully predicted SOI” and add a brief caveat that external validation was mixed (good on TGM, poor on GMB), likely due to range mismatch and storage effects.

2. Consider noting the short, two-time-point design to set expectations.

Introduction

Strengths: The motivation for using predator guts (diet + microbiome) as sentinel readouts is well framed; the ENSO context is clear; the gap (microbiome responses in skipjack) is articulated.

Suggestions: When attributing functional roles to higher-taxonomic ranks (e.g., family-level “probiotic” or immune associations), emphasize that function cannot be inferred directly from amplicon data and that families are ecologically broad. (You can preserve these links but frame them as hypotheses, supported by cited literature.)

Materials and Methods

Study design and sampling

Strengths: School-based design spanning a strong phase transition is compelling; within-school replication (∼10 fish) is practical.

Major points to address:

1. Temporal/Spatial confounding: All La Niña schools were sampled in December and all El Niño schools in March; locations span 05–09°S, 158–161°E. Please discuss potential spatial confounding and time-lag effects explicitly (ENSO signals propagate, and ecological responses can lag weeks–months). A map with school positions and dates would help.

2. Independence & hierarchical structure: Analyses sometimes operate at the individual level and sometimes at the school level (e.g., SOI models use school-level log10 abundances). Given the evident school effect in microbiome beta diversity (R² ≈ 0.15), please account for school as a grouping factor (random effect or blocking) wherever appropriate.

Laboratory and library prep

Strengths: Clear protocols; blocking primer for host COI; triplicate PCRs; negative controls at every stage; inclusion of ZymoBIOMICS community standards.

Suggestions: State explicitly whether any decontamination algorithm (e.g., decontam) was applied in addition to negative-control screening, especially relevant for families such as Propionibacteriaceae/Cutibacteriaceae that can be skin-associated. (You do note clean negatives.)

Bioinformatics and normalization

Strengths: DADA2, unambiguous phylum-level assignment, chloroplast/mitochondria removal, ANCOM-BC2 for differential abundance, and UniFrac for beta diversity—all standards.

Major points to address:

1. Alpha diversity computed on non-normalized counts: The text says alpha diversity indices were computed on non-normalized data. This can confound diversity with sequencing depth. Please either rarefy, include depth as a covariate, or use estimators robust to depth (e.g., breakaway/DivNet).

2. PERMANOVA assumptions: Before interpreting PERMANOVA, test multivariate dispersion (PERMDISP/betadisper). Unequal dispersion across groups (e.g., ENSO phases) can inflate Type I error. Please add the results or clarify whether they have already been provided.

3. Choice of distance & compositionality: You used unweighted UniFrac (presence/absence) after TSS normalization. Consider complementing with a compositional approach (Aitchison/CLR + Euclidean or PhILR) to ensure results are not an artifact of prevalence filtering or sequencing depth.

4. Multiple testing and effect sizes in ANCOM-BC2: Please report the false discovery rate (FDR) adjustment and the effect sizes, which are bias-corrected log-fold changes with confidence intervals (CIs), for the five significant families.

Environmental data

Suggestions: Specify the exact products for SST/CHLA (e.g., MODIS/VIIRS product IDs), temporal windows used for extraction (8-day composites are listed), and any spatial averaging buffers around coordinates. Consider testing time-lagged windows (e.g., 2–8 weeks) to match biological integration times.

Statistics and modeling

Major points to address:

1. Alpha-diversity GAMs vs. non-normality: You note indices are non-normal but then fit Gaussian GAMs to continuous covariates. Either transform, fit appropriate distributions, or use non-parametric smoothers with robust inference; report diagnostics.

2. SOI prediction with tiny n: The SOI models use five family-level predictors at the school level with n≈15—very small for random forests. Please conduct internal cross-validation that respects grouping (e.g., leave-one-school-out) and report those metrics. Current external validation is mixed (excellent for TGM; poor for GMB), consistent with overfitting and domain shift. Include permutation importance and partial-dependence plots.

Ethics: The submission form says, “All necessary permits … were obtained,” but PLOS ONE requires details for field research/animal work (permit numbers, issuing authority, and capture, handling, and euthanasia methods or rationale). Please add these in the Methods and Ethics Statement.

Results

Diet

Strengths include solid lab throughput, 16.3 million raw reads for diet analysis, 60 out of 75 samples containing non-Scombridae prey, and a dominance of fish species such as Myctophidae and Acanthuridae, with Cavoliniidae (pelagic gastropods) being particularly notable—this highlights the sensitivity of high-throughput sequencing (HTS). No prey family showed differences due to ENSO, which you interpret with caution.

Suggestions: Remind readers that read-based RA is semi-quantitative; you do use FOO and RA appropriately. Consider presenting Hill numbers and reporting sequencing depth summaries per sample alongside alpha metrics.

Microbiome

Strengths: 2.9 M post-QC reads across 142 samples; significant effects of ENSO/SOI on richness/evenness and beta diversity; clear visualization; Vibrionaceae dominance supports freshness.

Major points to address:

1. Add PERMDISP results; include effect sizes (e.g., R² with CIs) and 95% CIs/ribbons on GAM plots; annotate group n’s on figures.

2. Report ANCOM-BC2 statistics (adjusted p, LFC, CI) for the five families; indicate filtering thresholds and prevalence explicitly in the figure caption.

SOI prediction

Strength: Innovative use of microbiome to predict an environmental index; comparative modeling.

Concerns and requests:

1. Provide internal cross-validation that leaves out schools; with n≈15, report repeated CV variance.

2. Calibrate expectations in the text: training R²=0.991 and MSE=4.9 vs. poor performance on GMB suggests overfitting/domain mismatch; discuss implications for operational use.

Discussion and Conclusions

Strengths: Balanced interpretation; limitations acknowledged (limited timeframe, storage/time-range discrepancies in validation sets).

Suggestions:

1. Expand discussion of spatial/temporal confounding and school effects; emphasize that two time points cannot separate ENSO from seasonal or spatial gradients.

2. Reframe functional inferences as hypotheses (immune vs. “probiotic” roles) given family-level resolution; suggest future metagenomics/metatranscriptomics.

Figures, Tables, and Presentation

Positives: The heatmap and NMDS are clear; the feeding-strategy plot is helpful.

Edits:

1. Add sample-size annotations, CIs on GAM smooths, and color-blind-safe palettes.

2. Minor language edits: “drifting or anchored FADs” (not “of anchored”), “Actinopterygii” (spelling), and “were sequenced” (verb agreement).

Reviewer #2: This is a strong and timely study linking ENSO events with skipjack tuna diet and gut microbiomes. The dual use of diet and microbiome data, coupled with high-throughput sequencing and predictive modeling, makes the work novel and valuable. The paper is well structured, but it would benefit from clearer framing of limitations, more cautious interpretation of microbiome functions, and improved presentation.

1. Scope and Limitations

Highlight more clearly in the Discussion and Conclusion that the study covers only 15 schools over 4 months. Emphasize the need for longer-term, multi-year sampling.

2. Diet Analysis

In the Results, acknowledge the low proportion of successful prey detection (40%).

In the Discussion, expand on causes (empty stomachs, degraded DNA, host DNA dominance) and suggest solutions (alternative markers, host DNA depletion).

In the Conclusion, note that diet results are preliminary compared to microbiome findings.

3. Gut Microbiome

Add caution in the Discussion when linking bacterial families to immune or probiotic functions. Suggest future validation with metagenomics/metatranscriptomics.

4. Modeling

Expand in the Results/Discussion on why the Random Forest model failed with the GMB dataset (dataset differences, SOI range, storage issues).

Suggest adding environmental covariates and full ENSO cycle sampling for stronger predictive models.

5. Figures

Expand figure legends for clarity.

Add a workflow diagram (Methods) and a conceptual diagram (Discussion/Conclusion).

6. Methods & Ethics

Move detailed PCR conditions to Supplementary Material.

Add permit numbers in the Ethics Statement.

7. Language

Avoid repeating statistics in both Results and Discussion.

Correct minor grammar (e.g., “in study” → “in this study”) and standardize references.

.

Reviewer #1: **Yes:** Aratrika RayAratrika RayAratrika RayAratrika Ray

Reviewer #2: No

---

## [Author Response · Author response to Decision Letter 1]

8 Nov 2025

All responses below can also be found in the file "PlosOne_rebuttal_letter.docx".

Journal Requirements:

Author response:

We have reviewed and updated the manuscript to ensure compliance with PLOS ONE's style requirements, including proper file naming conventions, reference formatting, and figure/table formatting guidelines.

Author response:

We have added a complete ethics statement to the Methods section. As confirmed by our co-authors at the Pacific Community, no field access permits were required for this study, as all samples originated from commercial fishery catches.

This work was supported by the University of Canberra and by the European Union “Pacific-European-Union-Marine-Partnership” Programme (agreement FED/2018/397-941) grant to the Pacific Community. This publication was produced with the financial support of the European Union and Sweden. Its contents are the sole responsibility of the authors and do not necessarily reflect the views of the European Union and Sweden.

Author response:

We have included this statement in both cover letter and in the Funding sources section in the manuscript.

Author response:

We have added a complete ethics statement to the Methods section. The statement specifies that gut lining samples were collected from commercially caught skipjack tuna during routine operations, with no live animals handled for experimental purposes. All sampling followed standard commercial fishing practices and regional fisheries management protocols.

5. Please remove all personal information, ensure that the data shared are in accordance with participant consent, and re-upload a fully anonymized data set.

Author response:

We have reviewed all data files submitted with this manuscript to ensure no personal information is included. This study involves marine fish samples collected from commercial fisheries operations and does not involve human participants. We confirm that: (1) All spreadsheet files contain only scientific data (sample identifiers, fish measurements, date and locations, and environmental variables) with no personal information in any columns. (2) No columns are hidden in any data files. (3) File metadata (author, last modified by) has been cleared from all supplementary files. (4) The only names appearing in the manuscript are standard author citations in the reference list and necessary attributions for external datasets (GMB and TGM projects), which are required for scientific transparency and reproducibility.

Author response:

The reviewers did not recommend any specific citations to be added to the manuscript. All citations in the revised manuscript were selected based on their relevance to our study design, methodology, results, and interpretation.

Reviewer #1:

Title and Abstract

Strengths: Clear, accurate, and aligned with results; avoids overstating novelty. The abstract correctly contrasts weak diet signals with stronger microbiome responses, and it names the five key families and the modeling approach.

Suggestions:

1. Tone down “successfully predicted SOI” and add a brief caveat that external validation was mixed (good on TGM, poor on GMB), likely due to range mismatch and storage effects.

2. Consider noting the short, two-time-point design to set expectations.

Author response:

Thank you for recognizing the clarity and accuracy of our abstract and valuable suggestions. We have revised the abstract to address both suggestions.

Revised section:

Lines 32 – 37: A random forest model showed potential for ENSO phase classification based on the abundances of these five families, achieve high accuracy in internal validation, though the performance of external validation was mixed due to database differences . This study highlights the potential of skipjack tuna gut microbiome as indicators of rapid environmental changes, while acknowledging that the short sampling period requires longer-term validation across multiple ENSO cycles.

Introduction

Strengths: The motivation for using predator guts (diet + microbiome) as sentinel readouts is well framed; the ENSO context is clear; the gap (microbiome responses in skipjack) is articulated.

Suggestions: When attributing functional roles to higher-taxonomic ranks (e.g., family-level “probiotic” or immune associations), emphasize that function cannot be inferred directly from amplicon data and that families are ecologically broad. (You can preserve these links but frame them as hypotheses, supported by cited literature.)

Author response:

We have revised the Introduction to use more cautious language about gut microbiome functions, changing "is closely related to" to "has been associated with" when discussing relationships between microbiome and host physiology. Since the Introduction provides general context rather than discussing specific bacterial families, we reserved detailed discussion of family-level taxonomic limitations for the Discussion section where we interpret our microbiome results. Please refer to our response to Comment 2 in Discussion and Conclusions Section.

Revised section:

Lines 62 – 63: Second, the gut microbiome, which has been associated with the host’s nutritional provisioning, metabolism, and immune system functionality [20–25] …

Materials and Methods

Study design and sampling

Strengths: School-based design spanning a strong phase transition is compelling; within-school replication (∼10 fish) is practical.

Major points to address:

1. Temporal/Spatial confounding: All La Niña schools were sampled in December and all El Niño schools in March; locations span 05–09°S, 158–161°E. Please discuss potential spatial confounding and time-lag effects explicitly (ENSO signals propagate, and ecological responses can lag weeks–months). A map with school positions and dates would help.

Author response:

We acknowledge this important limitation and have added explicit discussion of potential confounding effects. However, several lines of evidence support that our observed microbiome patterns reflect direct ENSO effects rather than spatial artifacts: (1) The 4 - 5 degrees latitudinal sampling range is modest relative to skipjack tuna migration patterns and represents regional population variation rather than distinct stocks; (2) The absence of significant diet changes between sampling periods, despite clear microbiome shifts, suggests that gut microbiome responses to environmental change are more rapid and direct than traditional ecological indicators; (3) Among all environmental variables tested (SST, chlorophyll-a, SOI), only SOI significantly correlated with microbiome variation, supporting ENSO-specific rather than general temporal effects.

However, our two-timepoint design cannot definitively separate ENSO effects from seasonal variations. We have revised the manuscript to acknowledge this limitation and added a map showing sampling locations and dates as requested in the supplementary file (S1 Fig).

Revised section

Lines 107 – 110: While this design cannot entirely exclude potential confounding effects, the spatial range is modest relative to skipjack tuna migration patterns and represents regional rather than large scale variation [40, 41].

Lines 473 – 478: Spatial confounding might be less significant because the 4 - 5 degree latitudinal sampling range represents modest variation relative to skipjack tuna migration patterns and regional stock distribution [40,41,84]. However, we acknowledge that our two-timepoint design cannot definitively separate ENSO effects from seasonal variations. Systematic sampling across multiple ENSO cycles at fixed locations would be required to conclusively separate ENSO effects from temporal and spatial variation.

2. Independence & hierarchical structure: Analyses sometimes operate at the individual level and sometimes at the school level (e.g., SOI models use school-level log10 abundances). Given the evident school effect in microbiome beta diversity (R² ≈ 0.15), please account for school as a grouping factor (random effect or blocking) wherever appropriate.

Author response:

We acknowledge this important statistical consideration. Upon review, we found that school effects were appropriately handled in our analyses: (1) school was included as a factor in PERMANOVA tests, (2) Kruskal-Wallis tests for diversity indices inherently account for school-level grouping, and (3) SOI prediction models appropriately used school-level data.

Regarding the SOI prediction models specifically, we have re-examined our approach in light of this comment. Since SOI varies by sampling date, each school represents a unique SOI value, making school-level analysis the statistically appropriate approach. Our GAM and Random Forest models using school-level aggregated bacterial abundances to predict school-level SOI values are methodologically sound because both predictors and response variables operate at the same hierarchical level. Adding school as a random effect would be inappropriate in this context since SOI is inherently a school-level variable rather than an individual fish characteristic.

Revised section:

Lines 278 – 280: Bacterial family abundances were aggregated at the school level by calculating log10-transformed mean abundances, since SOI values correspond to sampling dates and locations that are consistent within each school.

Lines 400 – 401: A GAM and a RFM were constructed and compared, using the school-level log10-transformed abundance of the five families as predictor variables and SOI as the predicted variable.

Laboratory and library prep

Strengths: Clear protocols; blocking primer for host COI; triplicate PCRs; negative controls at every stage; inclusion of ZymoBIOMICS community standards.

Suggestions: State explicitly whether any decontamination algorithm (e.g., decontam) was applied in addition to negative-control screening, especially relevant for families such as Propionibacteriaceae/Cutibacteriaceae that can be skin-associated. (You do note clean negatives.)

Author response:

We appreciate this suggestion. We did not apply computational decontamination algorithms (such as decontam) beyond visual inspection of negative controls. Our negative controls showed no detectable DNA concentrations and were excluded from sequencing, indicating minimal contamination during sample processing. We acknowledge that families like Propionibacteriaceae can be skin-associated, but several factors support the validity of our detection: (1) samples were processed immediately after capture with sterile techniques, (2) negative controls from each processing stage showed no amplification, and (3) Propionibacteriaceae abundance patterns correlated specifically with environmental variables rather than showing random distribution that would suggest contamination artifacts.

However, we acknowledge that computational decontamination would strengthen future studies and have noted this in our revised Methods section.

Revised section:

Lines 296 – 297: No additional computational decontamination algorithms were applied beyond negative control screening.

Bioinformatics and normalization

Strengths: DADA2, unambiguous phylum-level assignment, chloroplast/mitochondria removal, ANCOM-BC2 for differential abundance, and UniFrac for beta diversity—all standards.

Major points to address:

1. Alpha diversity computed on non-normalized counts: The text says alpha diversity indices were computed on non-normalized data. This can confound diversity with sequencing depth. Please either rarefy, include depth as a covariate, or use estimators robust to depth (e.g., breakaway/DivNet).

Author response:

We appreciate this suggestion. We have revised our alpha diversity analysis by rarefying all samples to equal sequencing depth before calculating diversity indices. The rarefied analysis shows identical statistical significance patterns to our original analysis, confirming that our findings were not confounded by sequencing depth variation. We have updated our Methods section to describe the rarefaction procedure and revised Figure 3A and Supplementary File 2 accordingly.

Revised section:

Lines 223 – 226: Prior to alpha diversity analysis, samples were rarefied to equal sequencing depth (5019 reads per sample for diet data and 4017 reads per sample for microbiome data) using the phyloseq package (version 1.38.0) [51]. Shannon diversity [57], Chao1 species richness index [58], Simpson’s species evenness index [59] were calculated for rarefied diet and gut microbiome data using the microbiome package (version 1.23.1) [54].

2. PERMANOVA assumptions: Before interpreting PERMANOVA, test multivariate dispersion (PERMDISP/betadisper). Unequal dispersion across groups (e.g., ENSO phases) can inflate Type I error. Please add the results or clarify whether they have already been provided.

Author response:

We appreciate this suggestion to help improve the robustness of out result. We have conducted PERMDISP tests to verify PERMANOVA assumptions for all explanatory variables. The results show that our key findings are statistically valid: ENSO phases, SOI, and school all show equal multivariate dispersion (PERMDISP Pr > 0.1) while having significant PERMANOVA results (Pr < 0.001), confirming that these represent genuine centroid differences.

However, chlorophyll-a concentration and FADs showed both significant PERMDISP and PERMANOVA results, indicating potential dispersion artifacts rather than true centroid differences. All other variables showed non-significant PERMANOVA results. We have added all PERMDISP results to Supplementary File (S6 Table) and have updated our Methods and Results sections accordingly.

Revised section:

Lines 251 – 252: Before interpreting PERMANOVA results, PERMADIST tests were verified using vegan package (version 2.6.4) [64] to test for equal multivariate dispersion among groups.

Lines 354 – 360: Beta diversity of the gut microbiome was significantly affected by school (PERMANOVA test: F (14, 127) = 1.67, R2 = 0.16, Pr < 0.01,), ENSO (PERMANOVA test: F (1, 140) = 3.03, R2 = 0.02, Pr < 0.01), SOI

---

## [Decision Letter · Decision Letter 1]

3 Feb 2026

Dear Dr. Zhou,

We look forward to receiving your revised manuscript.

Kind regards,

Miquel Vall-llosera Camps

Senior Staff Editor

PLOS One

Journal Requirements:

Reviewers' comments:

Reviewer's Responses to Questions

**Comments to the Author**

Reviewer #1: (No Response)

Reviewer #2: All comments have been addressed

2. Is the manuscript technically sound, and do the data support the conclusions?

Reviewer #1: Partly

Reviewer #2: Yes

3. Has the statistical analysis been performed appropriately and rigorously?

Reviewer #1: Yes

Reviewer #2: Yes

4. Have the authors made all data underlying the findings in their manuscript fully available?

Reviewer #1: No

Reviewer #2: Yes

5. Is the manuscript presented in an intelligible fashion and written in standard English?

Reviewer #1: Yes

Reviewer #2: Yes

Reviewer #1: I thank the authors for their detailed and thoughtful responses to the initial review and for the substantial revisions made to the manuscript. Overall, the revised version shows a clear improvement in statistical rigour, transparency, and balanced interpretation. Many of the methodological and interpretative concerns raised in the first round have been carefully addressed, and I appreciate the effort invested in strengthening the analyses and presentation.

The authors deserve special recognition for their efforts:

(i) adding PERMDISP tests to support PERMANOVA interpretations;

(ii) incorporating a compositional (CLR-based) beta-diversity analysis alongside UniFrac;

(iii) revising alpha-diversity analyses to account for sequencing depth;

(iv) expanding ANCOM-BC2 reporting to include effect sizes and multiple-testing correction; and

(v) substantially improving the Discussion by acknowledging design limitations and reframing functional inferences more cautiously.

Most of my original comments have therefore been satisfactorily addressed. I have only a small number of remaining points, primarily related to clarity and consistency of interpretation rather than additional analyses.

1. Interpretation of microbiome vs. diet responses

The manuscript now appropriately emphasises the short, two-time-point sampling design and its limitations. However, in a few places, the text still implies that the absence of detectable diet changes, combined with microbiome shifts, provides evidence that gut microbiomes respond more rapidly or directly to ENSO forcing. While this interpretation is plausible, it cannot be demonstrated with the current design and remains inferential.

I recommend consistently framing this contrast as a hypothesis or inference, rather than as supporting evidence, throughout the Discussion. This would further align the interpretation with the acknowledged limitations of temporal and spatial confounding.

2. Treatment of school effects across analyses

The rationale for aggregating microbiome data at the school level for SOI modelling is statistically sound, and I agree that SOI is inherently a school-level variable. However, the manuscript alternates between describing school as a fixed effect and as a random effect in different parts of the Methods and Results.

For clarity, I suggest briefly harmonising this language and explicitly stating how school effects are handled in each analytical context (PERMANOVA, GAMs, and predictive models). This will help readers follow the hierarchical structure of the analyses more easily.

3. Predictive modeling and strength of claims

The additional leave-one-school-out cross-validation, permutation importance, and partial-dependence plots substantially strengthen the modelling section. The revised Discussion is also more balanced in acknowledging mixed external validation and domain-shift effects.

That said, given the minimal sample size (n ≈ 15 schools), repeated emphasis on “100% ENSO phase classification accuracy” risks being interpreted as stronger evidence of predictive power than is warranted. I recommend consistently qualifying these results as exploratory and proof-of-concept, rather than predictive in an operational sense, whenever they are mentioned.

4. Effect sizes and variance explained

Many microbiome–environment relationships explain a relatively small proportion of total variance (e.g., R² ≈ 0.02). This is not problematic and is common in ecological microbiome studies, but explicitly acknowledging this in the text would further strengthen the manuscript’s interpretive balance and prevent over-interpretation of statistically significant effects.

Overall assessment

In its current form, the manuscript presents a careful and methodologically sound proof-of-concept study demonstrating that skipjack tuna gut microbiome composition responds detectably to ENSO transitions under the conditions sampled, while diet signals remain comparatively weak. The authors have responded constructively to reviewer feedback, and the revised manuscript now aligns well with PLOS ONE’s emphasis on technical soundness and transparency.

Subject to the minor clarifications outlined above, I believe the manuscript is suitable for publication.

Recommendation: Minor revision.

Reviewer #2: The revisions have resolved all major concerns from the previous round, including statistical robustness, limitation acknowledgments, and balanced interpretations.

.

Reviewer #1: **Yes:** Aratrika Ray, PostDoc Researcher, Dept. of Aquatic Resources, Swedish University of Agricultural SciencesAratrika Ray, PostDoc Researcher, Dept. of Aquatic Resources, Swedish University of Agricultural SciencesAratrika Ray, PostDoc Researcher, Dept. of Aquatic Resources, Swedish University of Agricultural SciencesAratrika Ray, PostDoc Researcher, Dept. of Aquatic Resources, Swedish University of Agricultural Sciences

Reviewer #2: No

---

## [Author Response · Author response to Decision Letter 2]

2 Mar 2026

Reviewer #1:

I thank the authors for their detailed and thoughtful responses to the initial review and for the substantial revisions made to the manuscript. Overall, the revised version shows a clear improvement in statistical rigour, transparency, and balanced interpretation. Many of the methodological and interpretative concerns raised in the first round have been carefully addressed, and I appreciate the effort invested in strengthening the analyses and presentation.

The authors deserve special recognition for their efforts:

(i) adding PERMDISP tests to support PERMANOVA interpretations;

(ii) incorporating a compositional (CLR-based) beta-diversity analysis alongside UniFrac;

(iii) revising alpha-diversity analyses to account for sequencing depth;

(iv) expanding ANCOM-BC2 reporting to include effect sizes and multiple-testing correction; and

(v) substantially improving the Discussion by acknowledging design limitations and reframing functional inferences more cautiously.

Most of my original comments have therefore been satisfactorily addressed. I have only a small number of remaining points, primarily related to clarity and consistency of interpretation rather than additional analyses.

1. Interpretation of microbiome vs. diet responses

The manuscript now appropriately emphasises the short, two-time-point sampling design and its limitations. However, in a few places, the text still implies that the absence of detectable diet changes, combined with microbiome shifts, provides evidence that gut microbiomes respond more rapidly or directly to ENSO forcing. While this interpretation is plausible, it cannot be demonstrated with the current design and remains inferential.

I recommend consistently framing this contrast as a hypothesis or inference, rather than as supporting evidence, throughout the Discussion. This would further align the interpretation with the acknowledged limitations of temporal and spatial confounding.

Author response:

We thank the reviewer for this important clarification. We have revised the Discussion and Conclusions to consistently frame the microbiome-diet contrast as an inference rather than as supporting evidence.

Revised section:

Lines 460 – 461: In contrast to diet, the gut microbiome of skipjack tuna showed detectable response to environmental changes during the study period.

Lines 466- 467: This study may be the first to document detectable ENSO-associated changes in a marine fish gut microbiome over a four-month transitional period.

Lines 547 – 550: No significant relationship was found between diet composition and ENSO events potentially due to the high empty stomach rate, limited sample size, and the short sampling timeframe.

The gut microbiome showed detectable changes associated with ENSO events and SOI level, with significant changes in species richness and evenness.

2. Treatment of school effects across analyses

The rationale for aggregating microbiome data at the school level for SOI modelling is statistically sound, and I agree that SOI is inherently a school-level variable. However, the manuscript alternates between describing school as a fixed effect and as a random effect in different parts of the Methods and Results.

For clarity, I suggest briefly harmonising this language and explicitly stating how school effects are handled in each analytical context (PERMANOVA, GAMs, and predictive models). This will help readers follow the hierarchical structure of the analyses more easily.

Author response:

We thank the reviewer for this helpful suggestion. We have revised the Methods section to explicitly state how school is handled in each analysis type.

For PERMANOVA analysis, we added clarification that school is treated as a categorical fixed effect. For GAMs and predictive models, there are already school-handling methods, which we have retained.

Revised section:

Lines 251 – 253: Before interpreting PERMANOVA results, Permutational Multivariate Analysis of Dispersion (PERMDISP) tests were verified using vegan package (version 2.6.4) [64] to test for equal multivariate dispersion among groups, treating school as a categorical fixed effect.

3. Predictive modeling and strength of claims

The additional leave-one-school-out cross-validation, permutation importance, and partial-dependence plots substantially strengthen the modelling section. The revised Discussion is also more balanced in acknowledging mixed external validation and domain-shift effects.

That said, given the minimal sample size (n ≈ 15 schools), repeated emphasis on “100% ENSO phase classification accuracy” risks being interpreted as stronger evidence of predictive power than is warranted. I recommend consistently qualifying these results as exploratory and proof-of-concept, rather than predictive in an operational sense, whenever they are mentioned.

Author response:

We thank the reviewer for this important suggestion to ensure appropriate interpretation of our predictive modeling results. We have revised the manuscript to consistently qualify these results as exploratory and proof-of-concept throughout.

Revised section:

Lines 405 – 407: While the internal cross-validation for RFM showed moderate precision (R² = 0.90, RMSE = 5.2), it achieved perfect ENSO phase classification in this proof-of-concept analysis (100% accuracy, n = 15 schools) using the SOI threshold of 7 (S7 Table).

Lines 516 – 517: In this proof-of-concept study, internal cross-validation revealed that while SOI prediction showed moderate precision, the model achieved perfect accuracy in classifying ENSO phases. Similar classification performance was observed in external validation using TGM dataset, suggesting that this exploratory approach has the potential for further investigation with larger sample size across complete ENSO cycles.

Lines 537 – 538: Such comprehensive approaches would strengthen further operational deployment of microbiome-based environmental monitoring tools.

4. Effect sizes and variance explained

Many microbiome–environment relationships explain a relatively small proportion of total variance (e.g., R² ≈ 0.02). This is not problematic and is common in ecological microbiome studies, but explicitly acknowledging this in the text would further strengthen the manuscript’s interpretive balance and prevent over-interpretation of statistically significant effects.

Author response:

We thank the reviewer for this suggestion. We have added a clarifying statement to the Discussion section that contextualizes the R² values.

Revised section:

Lines 470 – 473: While ENSO and SOI explained a relatively small proportion of total variance in microbiome composition, such modest effect sizes are typical in ecological microbiome studies and can still reflect ecologically important responses to environmental change.

Reviewer #2:

The revisions have resolved all major concerns from the previous round, including statistical robustness, limitation acknowledgments, and balanced interpretations.

Author response:

We thank Reviewer #2 for their positive assessment and for confirming that our revisions have adequately addressed the concerns raised in the previous round. We appreciate their constructive feedback throughout the review process, which has substantially strengthened the manuscript.

---

## [Decision Letter · Decision Letter 2]

26 Mar 2026

Diet and gut microbiome of skipjack tuna (Katsuwonus pelamis) as indicators of environmental changes

PONE-D-25-46392R2

Dear Dr. Zhou,

We’re pleased to inform you that your manuscript has been judged scientifically suitable for publication and will be formally accepted for publication once it meets all outstanding technical requirements.

Kind regards,

Jianhong Zhou

Staff Editor

PLOS One

Additional Editor Comments (optional):

Reviewers' comments:

Reviewer's Responses to Questions

**Comments to the Author**

Reviewer #1: All comments have been addressed

Reviewer #2: All comments have been addressed

2. Is the manuscript technically sound, and do the data support the conclusions?

Reviewer #1: Yes

Reviewer #2: Yes

3. Has the statistical analysis been performed appropriately and rigorously?

Reviewer #1: Yes

Reviewer #2: Yes

4. Have the authors made all data underlying the findings in their manuscript fully available?

Reviewer #1: No

Reviewer #2: Yes

5. Is the manuscript presented in an intelligible fashion and written in standard English?

Reviewer #1: Yes

Reviewer #2: Yes

Reviewer #1: All the comments have been satisfactorily addressed by the authors, therefore I find its uitable now for acceptance and publication.

Reviewer #2: (No Response)

.

Reviewer #1: **Yes:** Aratrika Ray, Ph.D., Dept. of Aquatic Resources, Swedish University of Agricultural SciencesAratrika Ray, Ph.D., Dept. of Aquatic Resources, Swedish University of Agricultural SciencesAratrika Ray, Ph.D., Dept. of Aquatic Resources, Swedish University of Agricultural SciencesAratrika Ray, Ph.D., Dept. of Aquatic Resources, Swedish University of Agricultural Sciences

Reviewer #2: No

---

## [Editor Report · Acceptance letter]

PONE-D-25-46392R2

PLOS One

Dear Dr. Zhou,

I'm pleased to inform you that your manuscript has been deemed suitable for publication in PLOS One. Congratulations! Your manuscript is now being handed over to our production team.

Kind regards,

on behalf of

Dr. Jianhong Zhou

Staff Editor

PLOS One